# Training recurrent networks to generate hypotheses about how the brain solves hard navigation problems

**Ingmar Kanitscheider & Ila Fiete**
Department of Neuroscience
The University of Texas
Austin, TX 78712
`ikanitscheider, ilafiete @mail.clm.utexas.edu`

## Abstract

Self-localization during navigation with noisy sensors in an ambiguous world is computationally challenging, yet animals and humans excel at it. In robotics, *Simultaneous Location and Mapping* (SLAM) algorithms solve this problem through joint sequential probabilistic inference of their own coordinates and those of external spatial landmarks. We generate the first neural solution to the SLAM problem by training recurrent LSTM networks to perform a set of hard 2D navigation tasks that require generalization to completely novel trajectories and environments. Our goal is to make sense of how the diverse phenomenology in the brain's spatial navigation circuits is related to their function. We show that the hidden unit representations exhibit several key properties of hippocampal place cells, including stable tuning curves that remap between environments. Our result is also a proof of concept for end-to-end-learning of a SLAM algorithm using recurrent networks, and a demonstration of why this approach may have some advantages for robotic SLAM.

## 1   Introduction

Sensory noise and ambiguous spatial cues make self-localization during navigation computationally challenging. Errors in self-motion estimation cause rapid deterioration in localization performance, if localization is based simply on *path integration* (PI), the integration of self-motion signals. Spatial features in the world are often spatially extended (e.g. walls) or similar landmarks are found at multiple locations, and thus provide only partial position information. Worse, localizing in novel environments requires solving a chicken-or-egg problem: Since landmarks are not yet associated with coordinates, agents must learn landmark positions from PI (known as *mapping*), but PI location estimates drift rapidly and require correction from landmark coordinates.

Despite the computational difficulties, animals exhibit stable neural tuning in familiar and novel environments over several 10s of minutes [1, 2], even though the PI estimates in the same animals is estimated to deteriorate within a few minutes [3]. These experimental and computational findings suggest that the brain is solving some version of the *simultaneous localization and mapping* (SLAM) problem.

In robotics, the SLAM problem is solved by algorithms that approximate Bayes-optimal sequential probabilistic inference: at each step, a probability distribution over possible current locations and over the locations of all the landmarks is updated based on noisy motion and noisy, ambiguous landmark inputs [4]. These algorithms simultaneously update location and map estimates, effectively bootstrapping their way to better estimates of both. Quantitative studies of neural responses in rodents suggest that their brains might also perform high-quality sequential probabilistic fusion of motion and landmark cues during navigation [3]. The required probabilistic computations are difficult to

translate by hand into forms amenable to neural circuit dynamics, and it is entirely unknown how the brain might perform them.

We ask here how the brain might solve the SLAM problem. Instead of imposing heavy prior assumptions on the form a neural solution might take, we espouse a relatively model-free approach [5, 6, 7]: supervised training of recurrent neural networks to solve spatial localization in familiar and novel environments. A recurrent architecture is necessary because self-localization from motion inputs and different landmark encounters involves integration over time, which requires memory. We expect that the network will form representations of the latent variables essential to solving the task .

Unlike robotic SLAM algorithms that simultaneously acquire a representation of the agent's location and a detailed metric map of a novel environment, we primarily train the network to perform accurate localization; the map representation is only explicitly probed by asking the network to extract features to correctly classify the environment it is currently in. However, even if the goal is to merely localize in one of several environments, the network must have created and used a map of the environment to enable accurate localization with noisy PI. In turn, an algorithm that successfully solves the problem of accurate localization in novel environments can automatically solve the SLAM problem, as mapping a space then simply involves assigning correct coordinates to landmarks, walls, and other features in the space [4]. Our network solution exploits the fact that the SLAM problem can be considered as one of mapping sequences of ambiguous motion and landmark observations to locations, in a way that generalizes across trajectories and environments.

Our goal is to better understand how the brain solves such problems, by relating emergent responses in the trained network to those observed in the brain, and through this process to synthesize, from a function-driven perspective, the large body of phenomenology on the brain's spatial navigation circuits. Because we have access to all hidden units and control over test environments and trajectories, this approach allows us to predict the effective dimensionality of the dynamics required to solve the 2D SLAM task and make novel predictions about the representations the brain might construct to solve hard inference problems. Even from the perspective of well-studied robotic SLAM, this approach could allow for the learning and use of rich environment structure priors from past experience, which can enable faster map building in novel environments.

## 2 Methods

### 2.1 Environments and trajectories

We study the task of a simulated rat that must estimate its position (i.e., *localize* itself) while moving along a random trajectory in two-dimensional enclosures, similar to a typical task in which rats chase randomly scattered food pellets [8]. The enclosure is polygon-shaped and the rat does not have access to any local or distal spatial cues other than touch-based information upon contact with the boundaries of the environment (Figure 1A-B; for details see SI Text, section 1-4). We assume that the rat has access to noisy estimates of self-motion speed and direction, as might be derived from proprioceptive and vestibular cues (Figure 1A), and to boundary-contact information derived from its rare encounters with a boundary whose only feature is its geometry. On boundary contact, the rat receives information only about its distance and angle relative to the boundary (Figure 1B). This information is degenerate: it depends simply on the pose of the rat with respect to the boundary, and the same signal could arise at various locations along the boundary. Self-motion and boundary contact estimates are realistically noisy, with magnitudes based on work in [3].

### 2.2 Navigation tasks

We study the following navigation tasks:

- *Localization only: Localization in a single familiar environment*. The rat is familiar with the geometry of the environment but starts each trial at a random unknown location. To successfully solve the task, the rat must infer its location relative to a fixed point in the interior on the basis of successive boundary contacts and its knowledge of the environment's geometry, and be able to generalize this computation across novel random trajectories.
- *Generalized SLAM: Localization in novel environments*. Each trial takes place in a novel environment, sampled from a distribution of random polygons (Figure 1C; SI Text, section

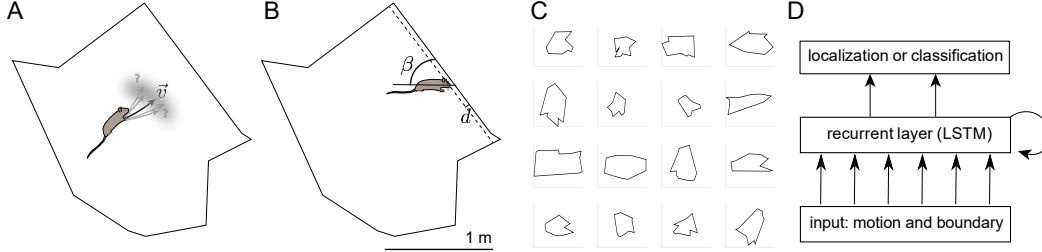

Figure 1: *Task setup.* Self-localization in 2D enclosures. *A* Noisy heading direction and speed inputs allow the simulated rat to update its location in the interior. *B* Occasional boundary contacts provide noisy estimates of the its relative angle ($\beta$) and distance ($d$) from the wall. Length scale is for localization-only task. *C* Samples from the distribution of random environments. *D* Architecture of the recurrent neural network.

1); the rat must accurately infer its location relative to the starting point by exploiting boundary inputs despite not knowing the geometry of its enclosure. To solve the task, the rat must be able to generalize its localization computations to trials with both novel trajectories and novel environments.

- *Specialized task: Localization in and classification of any of 100 familiar environments.* Each trial takes place in one of 100 known environments, sampled from a distribution of random polygons (Figure 1C; SI Text, section 1), but the rat does not know which one. The trial starts at a fixed point inside the polygon (known to rat through training), and the ongoing trajectory is random. In addition to the challenges of the localization tasks above, the rat must correctly classify the environment.

The environments are random polygons with 10 vertices. The center-to vertex lengths are drawn randomly from a distribution with mean 1m in the localization-only task or 0.33m in the specialized and generalized SLAM tasks.

## 2.3 Recurrent network architecture and training

The network has three layers: input, recurrent hidden and output layer (Figure 1D). The input layer encodes noisy self-motion cues like velocity and head direction change, as well as noisy boundary-contact information like relative angle and distance to boundary (SI Text, section 9). The recurrent layer contains 256 Long Short-Term Memory (LSTM) units with peepholes and forget gates [9], an architecture demonstrated to be able to learn dependencies across many timesteps [10]. We adapt the nonlinearity of the LSTM units to produce non-negative hidden activations in order to facilitate the comparison with neural firing rates[1]. Two self-localization units in the output perform a linear readout; their activations correspond to the estimated location coordinates. The cost function for localization is mean squared error. The classification output is implemented by a softmax layer with 100 neurons (1 per environment); the cost function is cross-entropy. When the network is trained to both localize and classify, the relative weight is tuned such that the classification cost is half of the localization cost. Independent trials used for training: 5000 trials in the localization-only task, 250,000 trials in the specialized task, and 300,000 trials in the generalized task. The network is trained using the Adam algorithm [11], a form of stochastic gradient descent. Gradients are clipped

$$
\begin{aligned}
i_t &= \sigma(W_{xi}x_t + W_{hi}h_{t-1} + w_{ci} \odot c_{t-1} + b_i) \\
f_t &= \sigma(W_{xf}x_t + W_{hf}h_{t-1} + w_{cf} \odot c_{t-1} + b_f) \\
c_t &= f_t c_{t-1} + i_t \tanh(W_{xc}x_t + W_{hc}h_{t-1} + b_c) \\
o_t &= \sigma(W_{xo}x_t + W_{ho}h_{t-1} + w_{co} \odot c_t + b_o) \\
h_t &= o_t \tanh([c_t]_+)
\end{aligned}
$$

where $\sigma$ is the logistic sigmoid function, $h$ is the hidden activation vector, $i$, $f$, $o$ and $c$ are respectively the input gate, forget gate, output gate and cell activation vectors, $a \odot b$ denotes point-wise multiplication and $[x]_+$ denotes rectification.

to 1. During training performance is monitored on a validation set of 1000 independent trials, and network parameters with the smallest validation error are selected. All results are cross-validated on a separate set of 1000 test trials to ensure the network indeed generalizes across new random trajectories and/or environments.

# 3    Results

## 3.1    Network performance on spatial tasks rivals optimal performance

### 3.1.1    Localization in a familiar environment

The trained network, starting a trial from an unknown random initial position and running along a new random trajectory, quickly localizes itself within the space (Figure 2, red curve). The mean location error (averaged over new test trials) drops as a function of time in each trial, as the rat encounters more boundaries in the environment. After about 5 boundary contacts, the initial error has sharply declined.

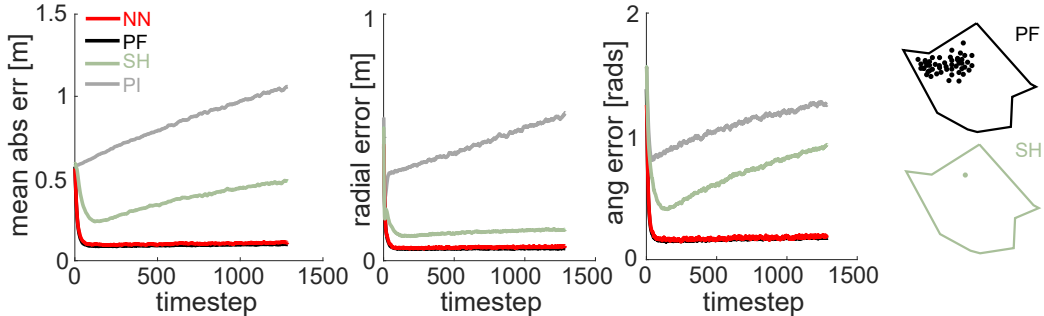

Figure 2: *Localization in a single familiar environment.* Mean absolute error on the localization-only task (left), radial error measured from origin (middle) and angular error (right). One time step corresponds to 0.77 seconds. Network performance (red, NN) is compared to that of the particle filter (black, PF). Also shown: single hypothesis filter (light red, SH) and simple path integration (gray, PI) estimates as controls.

The drop in error over time and the final error of the network match that of the optimal Bayesian estimator with access to the same noisy sensory data but perfect knowledge of the boundary coordinates (Figure 2, black). The optimal Bayesian estimator is implemented as a *particle filter* (PF) with 1000 particles and performs fully probabilistic sequential inference about position, using the environment coordinates and the noisy sensory data. The posterior location distributions are frequently elongated in an angular arc and multimodal (thus far from Gaussian).

Both network and PF vastly outperform pure PI. First, since the PI estimate does not have access to boundary information, it cannot overcome initial localization uncertainty due to the unknown starting point. Second, the error in the PI estimate of location grows unbounded with time, as expected due to the accumulating effects of noise in the motion estimates (Figure 2, gray). In contrast, the errors in the network and PF – which make use of the same motion estimates – remain bounded.

Finally we contrast the performance of the network and PF with the *single hypothesis* (SH) algorithm, which updates a single location estimate (rather than a probability distribution) by taking into account motion, contact, and arena shape. The SH algorithm can be thought of as an abstraction of neural bump attractor models [12, 13], in which an activity bump is updated using PI and corrected when a landmark or boundary with known spatial coordinates is observed. The SH algorithm overcomes, to a certain degree, the initial localization uncertainty due to the unknown starting position, but the error steadily increases thereafter. It still vastly underperforms the network and PF, since it is not able to efficiently resolve the complex-shaped uncertainties induced by featureless boundaries.

### 3.1.2 Localization in novel environments

The network is trained to localize within a different environment in each trial, then tested on a set of trials in different novel environments.

Strikingly, the network localizes well in the novel environments, despite its ignorance about their specific geometry (Figure 3A, red). While the network (unsurprisingly) does not match the performance of an *oracular* PF that is supplied with the arena geometry at the beginning of the trial (Figure 3A, black), its error exceeds the oracular PF by only $\approx 50\%$, and it vastly outperforms PI-based estimation (Figure 3A, gray) and a naive Bayesian (NB) approach that takes into account the distribution of locations across the ensemble of environments (Figure 3A, reddish-gray; SI section 8).

Compared to robotic SLAM in open-field environments, this task setting is especially difficult since distant boundary information is gathered only from sparse contacts, rather than spatially extended and continuous measurements with laser or radar scanners.

### 3.1.3 Localization in and classification of 100 familiar environments

The network is trained on 100 environments then tested in an arbitrary environment from that set. The goal is to identify the environment and localize within it, from a known starting location. Localization initially deteriorates because of PI errors (Figure 3B, red). After a few boundary encounters, the network correctly identifies the environment (Figure 3C), and simultaneously, localization error drops as the network now associates the boundary with coordinates for the appropriate environment. The network's localization error post-classification matches that of an oracular PF with full knowledge about the environment geometry. Within 200s of exploration within the environment, classification performance is close to 100%.

As a measure of the efficacy of the neural network in solving the specialized task, we compare its performance to PFs that do not know the identity of the environment at the outset of the trial (PF SLAM) and that perform both localization and classification, with varying numbers of particles, Figure 3D-E. For classification, the asymptotic network performance with 256 recurrent units is comparable to a 10,000 particle PF SLAM, while for localization, the asymptotic network performance is comparable to a 4,000 particle PF SLAM, suggesting that the network is extremely efficient. Even the 10,000 particle PF SLAM classification estimate sometimes prematurely collapses to not always the correct value. The network is slower to select a classification, and is more accurate, improving on a common problem with particle-filter based SLAM caused by particle depletion.

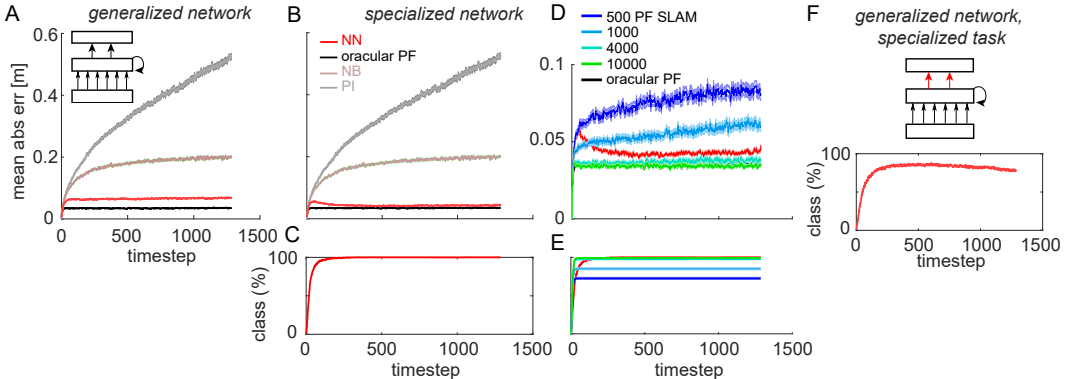

Figure 3: *Localization and classification in the generalized and specialized SLAM tasks. A* Localization performance of the generalized network (red, NN) tested in novel environments, compared to a PF that knows the environment identity (black, oracular PF). Controls: PI only (gray, PI) and a naive Bayes filter (see text and SI; reddish-gray, NB). *B* Same as (A), but for the specialized network tested in 100 familiar environments. *C* Classification performance of the specialized network in 100 familiar environments. *D-E* Localization and classification by a SLAM PF with different number of particles, compared to the specialized network in 100 familiar environments. *F* Classification performance of the general network after retraining of the readout weights on the specialized task.

### 3.1.4 Spontaneous classification of novel environments

In robotic SLAM, algorithms that self-localize accurately in novel environments in the presence of noise must simultaneously build a map of the environments. Since the network in the general task in Figure 3A successfully localizes in novel environments, and is able to distinguish between them though they are quite similar, we conjecture that it must entertain a spontaneous representation of the environment.

To test this hypothesis we fix the input and recurrent weights of the network trained on the generalized task (completely novel environments) and retrain it on the specialized task (one out of hundred familiar environments), whereby only the readout weights are trained for classification. We find that the classification performance late in each trial is close to 80%, much higher than chance (1%), Figure 3F. This implies that the hidden neurons spontaneously build a representation that separates novel environments so they can be linearly classified. This separation can be interpreted as a simple form of spontaneous map-building. However, this spontaneous map-building is done with fixed weights - this is different than standard Hopfield-type network models that require synaptic plasticity to learn a new environment.

## 3.2 Comparison with and predictions for neural representation

Neural activity in the hippocampus and entorhinal cortex – areas involved in spatial navigation – has been extensively catalogued, usually while animals chase randomly dropped food pellets in open field environments. It is not always clear what function the observed responses play in solving hard navigation problems, or why certain responses exist. Here we compare the responses of our network, which is trained to solve such tasks, with the experimental phenomenology.

Hidden units in our network exhibit stable place tuning, similar to place cells in CA1/CA3 of the hippocampus [14, 15, 16], Figure 4A,B (left two columns). Stable place fields are observed across tasks – the network trained to localize in a single familiar environment exhibits stable fields there, while the networks trained on the specialized and generalized tasks exhibit repeatedly stable fields in all tested environments.

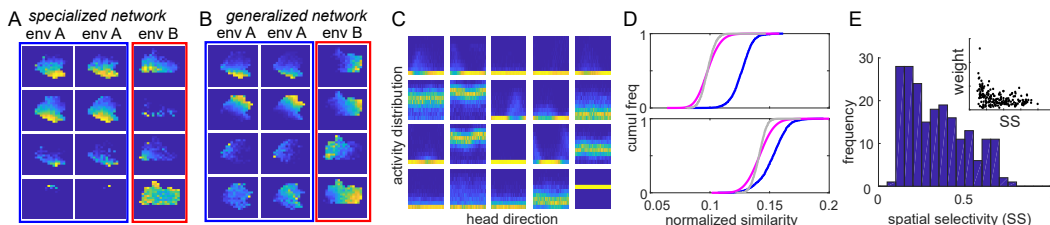

Figure 4: *Neuron-like representations. A* Spatial tuning of four typical hidden units from the specialized network, measured twice with different trajectories in the same environment (columns 1-2, blue box). The same cells are measured in a second environment (column 3, red box). *B* Same as *A* but for the generalized network; both environments were not in the training set. *C* Hidden units (representative sample of 20) are not tuned to head direction. *D* Cumulative distribution of similarity of hidden unit states in the specialized (top) and generalized (bottom) networks, for trials in the same environment (blue) versus trials in different environments (pink). Control: similarity after randomizing over environments (gray). *E* Spatial selectivities of hidden units in the specialized network. Inset: spatial selectivity (averaged across environments) versus effective projection strength to classifier neurons, per hidden unit.

The hidden units, all of which receive head direction inputs and use this data to compute location estimates, nevertheless exhibit weak to nil head direction tuning, Figure 4C, again similar to observations in rodent place cells [17] (but see [18] for a report of head direction tuning in bat place cells).

Between different environments, the network trained on the specialized task exhibits clear *remapping* [19, 20], both global and local: cells fire in some environments and not others, and cells that were co-active in one environment are not in another, Figure 4A,B (third column). There is, in addition, a substantial amount of rate modulation in cells when they do not globally remap. Strikingly, the network trained on the generalized task exhibits different but stable and reproducible maps of different

novel environments *with remapping*, even though the input and recurrent connections were never readjusted for these novel environments, Figure 4B. This result suggests a computation that is distinct from the dynamics of settling into pre-trained fixed maps for different environments.

The similarity and dissimilarity of the representations within the same environment and across environments, in the specialized and generalized tasks are quantified in Figure 4D: the representations are randomized across environments but stable within an environment.

For networks trained on the specialized or generalized tasks, the spatial selectivity of hidden units in an environment - measured as the fraction of the variance of each hidden neuron's activation that can be explained by location - is broad and long-tailed or sparse, Figure 4E: a few cells exhibit high selectivity, many have low selectivity. Interestingly, cells with low spatial selectivity in one environment also tend to have low selectivity across environments (in other words, the distribution in selectivity per cell across environments is narrower than the distribution of selectivity across cells per environment). Indeed, spatial information in hippocampal neurons seems to be concentrated in a small set of neurons [21], an experimental observation that seemed to run counter to the information-theoretic view that whitened representations are most efficient. However, our 256-neuron recurrent network, which efficiently solves a hard task that requires $10^4$ particles, seems to do the same.

There is a negative correlation between spatial selectivity and the strength of feedforward connections to the classification units: Hidden units that more strongly drive classification also tend to be less spatially selective, Figure 4E (inset). In other words, some low spatial selectivity cells correspond to what are termed *context* cells [22]. It remains unclear and the focus of future work to understand the role of the remaining cells with low spatial selectivity.

## 3.3   Inner workings of the network

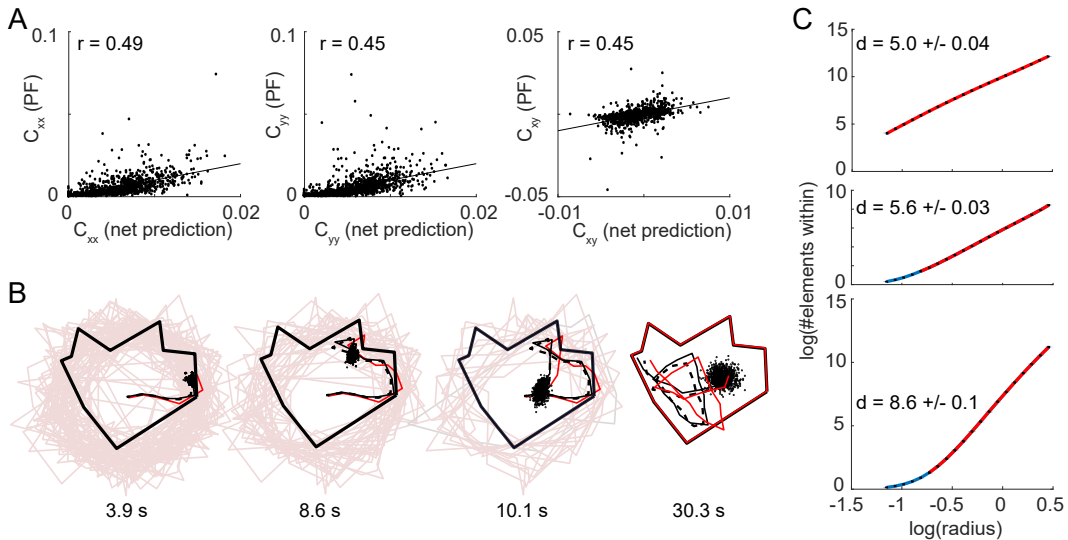

Figure 5: *Inner workings of the network A* Hidden units in the localization-only network predict the covariances ($C_{xx}, C_{yy}, C_{xy}$) of the posterior location $(x, y)$ distributions in the particle filter. *B* Light red: snapshots of the narrowing set of potential environment classifications by the specialized neural network at different early times in a trajectory, as determined by the activation of classifier neurons in the output layer. *C* Dimensionality of the hidden representations, estimated by the correlation dimension measure: localization network (top), specialized network (middle), generalized network (bottom). Dimensionality estimated from across-environment pooled responses for the latter two networks.

Beyond the similarities between representations in our hidden units and neural representations, what can we learn about how the network solves the SLAM problem?

The performance of the network compared to the particle filter (and its superiority to simpler strategies used as controls) already implies that the network is performing sophisticated probabilistic computations about location. If it is indeed tracking probabilities, it should be possible to predict the

uncertainties in location estimation from the hidden units. Indeed, all three covariance components related to the location estimate of the particle filter can be predicted by cross-validated linear regression from the hidden units in the localization-only network (Figure 5A).

When first placed into one of 100 familiar environments, the specialized network simultaneously entertains multiple possibilities for environment identity, Figure 5B. The activations of neurons in the soft-max classification layer may be viewed as a posterior distribution over environment identity. With continued exploration and boundary encounters, the represented possibilities shrink until the network has identified the correct environment.

Unlike the particle filter and contrary to neural models that implement probabilistic inference by stochastic sampling of the underlying distribution [23], this network implements ongoing near-optimal probabilistic location estimation through a fully deterministic dynamics.

Location in 2D spaces is a continuous 2D metric variable, so one might expect location representations to lie on a low-dimensional manifold. On the other hand, SLAM also involves the representation of landmark and boundary coordinates and the capability to classify environments, which may greatly expand the effective dimension of a system solving the problem. We analyze the fractal manifold dimension of the hidden layer activities in the three networks, Figure 5C[2]. The localization-only network has a dimension $D = 5.0$. Surprisingly, the specialized network states (pooled across all 100 environments) are equally low-dimensional: $D = 5.6$. The generalized network states, pooled across environments, have dimension $D = 8.6$. (The dimensionality of activity in the latter two networks, considered in single environments only, remains the same as when pooled across environments.) This implies that the network extracts and representing only the most relevant summary statistics required to solve the 2D localization tasks, and that these statistics have fairly low dimension. These dimension estimates could serve as a prediction for hippocampal dynamics in the brain.

## 4 Discussion

By training a recurrent network on a range of challenging navigation tasks, we have generated – to our knowledge – the first fully neural SLAM solution that is as effective as particle filter-based implementations. Existing neurally-inspired SLAM algorithms such as RatSLAM [24] have combined attractor models with semi-metric topological maps, but only the former was neurally implemented. [25] trained a bidirectional LSTM network to transform laser range sensor data into location estimates, but the network was not shown to generalize across environments. In contrast, our recurrent network implementation is fully neural and generalizes successfully across environments with very different shapes. (Also see [26], a new preprint posted while this paper was under review, reporting on a SLAM implementation with recurrent neural network components. Other recent efforts to combine DNNs with SLAM usually apply DNNs to the input visual input, and the outputs of the DNN are then fed into an existing SLAM algorithm [27, 28]. By contrast, our focus has been on finding neural solutions to the SLAM algorithm itself.)

Previous hand-designed models such as the multichart attractor model of Samsonovich & Mc-Naughton [12] could path integrate and use landmark information to correct the network's PI estimate in many different environments. Yet our model substantially transcends those computational capabilities: First, our model performs sequential probabilistic inference, not simply a hard resetting of the PI estimate according to external cues. Second, our network reliably localizes in 100 environments with 256 LSTM units (which corresponds to 512 dynamical units); the low capacity of the multichart attractor model would require about 175,000 neurons for the same number of environments. This comparison suggests that the special network architecture of the LSTM not only affects learnability, but also capacity. Finally, unlike the multichart attractor model, our model is able to linearly separate completely novel environments without changing its weights, as shown in section 3.1.4.

Despite its success in reproducing some key elements of the phenomenology of the hippocampus, our network model does not incorporate many biological constraints. This is in itself interesting, since it suggests that observed phenomena like stable place fields and remapping may emerge from the computational demands of hard navigation tasks rather than from detailed biological

constraints. It will be interesting to see whether incorporating constraints like Dale's law and the known gross architecture of the hippocampal circuit results in the emergence of additional features associated with the brain's navigation circuits, such as sparse population activity, directionality in place representations in 1D environments, and grid cell-like responses.

The choice of an LSTM architecture for the hidden layer units, involving multiplicative input, output and forget gates and persistent cells, was primarily motivated by its ability to learn longer time-dependencies. One might wonder whether such multiplicative interactions could be implemented in biological neurons. A model by [29] proposed that dendrites of granule cells in the dental gyrus contextually gate projections from grid cells in the entorhinal cortex to place cells. Similarly, granule cells could implement LSTM gates by modulating recurrent connections between pyramidal neurons in hippocampal area CA3. LSTM cells might be interpreted as neural activity or as synaptic weights updated by a form of synaptic plasticity.

The learning of synaptic weights by gradient descent does not map well to biologically plausible synaptic plasticity rules, and such learning is slow, requiring a vast number of supervised training examples. Our present results offer a hint that, through extensive learning, the generalized network acquires useful general prior knowledge about the structure of natural navigation tasks, which it then uses to map and localize in novel environments with minimal further learning. One could thus argue that the slow phase of learning is evolutionary, while learning during a lifetime can be brief and driven by relatively little experience in new environments. At the same time, progress in biologically plausible learning may one day bridge the efficiency gap to gradient descent [30].

Finally, although our work is focused on understanding the phenomenology of navigation circuits in the brain, it might also be of some interest for robotic SLAM. SLAM algorithms are sometimes augmented by feedforward convolutional networks to assist in specific tasks like place recognition (see e.g. [27, 28]) from camera images, but the geometric calculations and parameters at the core of SLAM algorithms are still largely hand-specified. By contrast, this work provides a proof of concept for the feasibility end-to-end learning of SLAM algorithms using recurrent neural networks and shows that the trained network provides a solution to the particle depletion problem that plagues many particle filter-based approaches to SLAM and is highly effective in identifying which low-dimensional summary statistics to update over time.

## Acknowledgments

This work is supported by the NSF (CRCNS 26-1004-04xx), an HFSP award to IRF (26-6302-87), and the Simons Foundation through the Simons Collaboration on the Global Brain. The authors acknowledge the Texas Advanced Computing Center (TACC) at The University of Texas at Austin (URL: http://www.tacc.utexas.edu) for providing HPC resources that have contributed to the research results reported within this paper.

## Footnotes

[1]The LSTM equations are implemented by the equations:

[2]To estimate the fractal dimension, we use "correlation dimension": measure the number of states across trials that fall into a ball of radius $r$ around a point in state space. The slope of $\log(\#\text{states})$ versus $\log(r)$ is the fractal dimension at that point.

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
