[Reviews · NeurIPS 2017]

Reviewer 1



An LSTM recurrent neural network is trained to perform simultaneous localization and mapping (SLAM) tasks, given noisy odometry data and occasional input from contacts with walls. The paper is well written and I find it interesting to study recurrent neural network solutions of SLAM. But the idea of using LSTM-RNNs for SLAM problems is not new (as the authors mention in the discussion, line 256) and the results do not really surprise me. Given the amount of training trials, the network has presumably seen sufficiently many trajectories (and variations of the environment) to perform well on most test examples (and if the difference between training and testing error is too large, follow the standard recipes and tune the number of hidden neurons and the number of training trials). A more interesting result would be it the LSTM-RNN would generalize to out of distribution samples, i.e. if its performance would be comparable to that of a PF SLAM in totally different environments than the ones seen during training. It is also not clear to me, what we learn about neural recordings. Given that the task of the specialized networks is to output the position (independent of head direction) and the identity of the current environment (independent of current location) it does not suprise to see head-direction independent tuning and cells with low spatial selectivity. More comments and questions: 1. line 5: "We generate the first neural solution to the SLAM problem": This sounds too strong given papers like Blum and Abbott 1996 http://dx.doi.org/10.1162/neco.1996.8.1.85 Foster et al. 2000 dx.doi.org/10.1002/(SICI)1098-1063(2000)10:1 < 1::AID-HIPO1 > 3.0.CO;2-1 or the one already cited by Förster et al. 2007 2. line 46-48: I don't see why the network must have created a map. Isn't it sufficient to extract some features that identify each environment? 3. The font size in the figures is a bit too small for my taste; especially Figure 3. 4. Caption 4D: I guess it should read "specialized (top) and generalized (bottom)" 5. line 208: How is the spatial selectivity measured? 6. Figure 5B: How is the time in seconds related to the (discrete) simulation steps? 7. line 242: What is the fractal manifold dimension? 8. Suppl. line 19: I was curious how rare the events are actually. Is it something like on average every 10th step for the small environments (task 2 and 3) and every 30th step for the larger environments? 9. Suppl. line 101: Why do the tiles extend up to 35 cm? Isn't it extremely unlikely to trigger a boundary event with this distance? ====== I read the author’s rebuttal and my colleagues’ review. My main concerns remain: 1. A powerfull off-the-shelf function approximator (LSTM) is applied to artificial data. This is a setting where one can always achieve a small test error. One can play with the difficulty of the task, the amount of data and the meta-parameters of the function approximator. It would be a different thing if tested on completely different environments where the network would truely have to learn to solve the SLAM problem independently of the environment. 2. As mentioned above, given the tasks it does not surprise to see head-direction independent tuning and cells with low spatial selectivity. Any model that solves these tasks has to represent head-direction independent and position independent information. It is not clear to me what we learn on top of this with this specific model.

Reviewer 2



The authors train a recurrent neural network with LSTMs to perform navigation tasks. The network is provided with the speed and heading direction of the simulated rat, along with boundary events. Using this information, the network has to estimate the current position of the rat, and (in some cases) the identity of the environment. The network is able to perform the tasks, with an accuracy similar to that of a particle filter. Neurons in the network exhibit spatial selectivity and remapping. This is an important and impressive work. Unlike vision, there are very few works using end to end learning of navigation tasks. This is despite the fact that there is ample biological information on the neural activity during navigation, and it has many real world applications (e.g. robotics). The framework used allows the authors to study both the recognition of an environment and navigation within it. This is particularly important when contrasting with the usual neuroscience experiments in which the environments are very limited. Real world environments are, of course, much more variable, but this framework can chart a path towards their study. Limitations and comments: 1. The trajectory is randomly generated, as opposed to an actual rat that does not necessarily care about its absolute position, but rather on where it has to go. 2. The environments are different, but not very different. The failure of the single hypothesis filter indicates that the variability is enough to affect navigation. But it is unclear whether the network is able to operate when the environments vary greatly in size (implying multiple timescales of path integration). 3. Fig 3 is impressive, but NN is comparable to 4000, not 10000. Does the PF SLAM have a similar demand (50% of the loss) on classification as the NN? If not, then perhaps perfect classification is not needed to localize well because some environments are similar 4. Figure 4D legend – should be top/bottom, not left/right 5. Figure 5A – marking the diagonal would greatly assist here 6. Figure 2,3 – the choice of colors is not very good. It’s hard to discern the light red.

Reviewer 3



In this study, the authors trained recurrent neural network models with LSTM units to solve a well studied problem, namely Simultaneous Location and Mapping (SLAM). The authors trained the network to solve several different version of the problems, and made some qualitative comparison to rodent physiology regard spatial navigation behavior. In general, I think the authors' approach follows an interesting idea, namely using recurrent network to study the cognitive functions of the brain. This general idea has recently started to attract substantial attentions in computational neuroscience. The authors tested several version of the localization tasks, which is extensive. And the attempt to compare the model performance to the neurophysiology should be appreciated. They also attempted to study the inner working of the training network by looking into the dimensionality of the network, which reveals a relatively low dimensional representation, but higher than the dimension of the physical space. Concerns- The algorithms used is standard in machine learning, thus my understanding is that the main contribution of the paper presumably should come from either solving the SLAM problem better or shed some light from the neuroscience perspective. However, I am not sure about either of them. 1) For solving the SLAM problem, I don't see a comparison to the state-of-the-art algorithms in SLAM, so it is unclear where the performance of this algorithm stands. After reading through other reviewers' comments and the rebuttal, I share similar concerns with Reviewer 1- it is unclear whether the networks learn to solve the SLAM generically, and whether the network can perform well in completely different environments. The set-up the authors assumed seem to be restrictive, and it is unclear whether it can apply to realist SLAM problem. I agree with Reviewer 1 that this represents one important limitation of this study I initially overlooked, and it was not satisfyingly addressed in the rebuttal. 2) From the neuroscience perspective, some of the comparison the authors made in Section 3.2 are potentially interesting. However, my main concern is that the "neuron-like representations" shown in Fig.4A,B are not really neuron-like according to the rodent physiology literature. In particular, the place fields shown in Fig4A and 4B do not resemble the place fields reported in rodent physiology, which typically have roughly circular symmetric shape. I'd certainly appreciate these results better if these spatial tuning resembles more to the known rodent physiology. Can the authors show the place fields of all the neurons in the trained network, something like Fig 1 in Wilson & McNaughton (1993, Science)? That'd be helpful in judging the similarity of the representation in the trained network and the neural representation of location in rodents. Also, in the rodent literature, the global remapping and rate remapping are often distinguished and the cause for these two types of remapping are not well-understood at this point. Here the authors only focus on global remapping. One potentially interesting question is whether the rate remapping also exist in the trained network. The paper would be stronger if the authors could shed some light on the mechanisms for these two types of remapping from their model. Just finding global remapping across different environment isn't that surprising in my view. Related to Fig4E, it is known that neurons in hippocampus CA1 exhibit a heavy tail in their firing rate (e.g, discussed in ref[20]). Do the authors see similar distributions of the activity levels in the trained network? 3) The manuscript is likely to be stronger if the authors could emphasize either the performance of the network in performing SLAM or explaining the neurophysiology. Right now, it feels that the paper puts about equal weights on both, but neither part is strong enough. According to the introduction and abstract, it seems that the authors want to emphasize the similarity to the rodent physiology. In that regard, Fig. 2 and Fig. 3 are thus not particularly informative, unless the authors show the performance from the recurrent network is similar to the rodent (or maybe even humans, if possible) behavior in some interesting ways. To emphasize the relevance to neuroscience, it is useful to have a more extended and more thorough comparison to the hippocampus neurophysiology, though I am not sure if that's possible given the place fields of the trained network units do not resemble the neurophysiology that well, as I discussed earlier. Alternatively, to emphasize the computational power of the RNN model in solving SLAM, it would be desirable to compare the RNN to some state-of-the-art algorithms. I don't think the particle filtering approach the authors implemented represents the state-of-the-art in solving SLAM. But I could be wrong on this, as I am not very familiar with that literature. Finally, a somewhat minor point- the authors make the comparison on the capacity of the network to the multi-chart attractor model in the Discussion, but I am not sure how that represents a fair comparison. I'd think one LSTM unit is computational more powerful than a model neuron in a multi-chart attractor model. I'd like to understand why the authors think a comparison can be drawn here.